# Physiological Response of Lettuce (*Lactuca sativa* L.) Grown on Technosols Designed for Soil Remediation

**DOI:** 10.3390/plants13223222

**Published:** 2024-11-16

**Authors:** Mateo González-Quero, Antonio Aguilar-Garrido, Mario Paniagua-López, Carmen García-Huertas, Manuel Sierra-Aragón, Begoña Blasco

**Affiliations:** 1Department of Plant Physiology, Faculty of Sciences, University of Granada, Av. de Fuente Nueva s/n, 18071 Granada, Spain; matgonque@gmail.com (M.G.-Q.); cgarciahuertas@correo.ugr.es (C.G.-H.); bblasco@ugr.es (B.B.); 2Department of Soil Science and Agricultural Chemistry, Faculty of Sciences, University of Granada, Av. de Fuente Nueva s/n, 18071 Granada, Spain; mpaniagua@ugr.es (M.P.-L.); msierra@ugr.es (M.S.-A.)

**Keywords:** PHEs toxicity, soil properties, plant growth, photosynthesis, oxidative stress, tolerance mechanisms, reactive oxygen species, malondialdehyde, proline, ascorbate-glutathione cycle

## Abstract

This study focuses on the physiological response of lettuce grown on Technosols designed for the remediation of soils polluted by potentially harmful elements (PHEs: As, Cd, Cu, Fe, Pb, and Zn). Lettuce plants were grown in five treatments: recovered (RS) and polluted soil (PS) as controls, and three Technosols (TO, TS, and TV) consisting of 60% PS mixed with 2% iron sludge, 20% marble sludge, and 18% organic wastes (TO: composted olive waste, TS: composted sewage sludge, and TV: vermicompost of garden waste). The main soil properties and PHE solubility were measured, together with physiological parameters related to phytotoxicity in lettuce such as growth, photosynthetic capacity, oxidative stress, and antioxidant defence. All Technosols improved unfavourable conditions of PS (i.e., neutralised acidity and enhanced OC content), leading to a significant decrease in Cd, Cu, and Zn mobility. Nevertheless, TV was the most effective as the reduction in PHEs mobility was higher. Furthermore, lettuce grown on TV and TO showed higher growth (+90% and +41%) than PS, while no increase in TS. However, lower oxidative stress and impact on photosynthetic rate occurred in all Technosols compared to PS (+344% TV, +157% TO, and +194% TS). This physiological response of lettuce proves that PHE phytotoxicity is reduced by Technosols. Thus, this ecotechnology constitutes a potential solution for soil remediation, with effectiveness of Technosols depending largely on its components.

## 1. Introduction

Soil pollution by potentially harmful elements (PHEs) poses a major risk to ecosystems and public health, with more than five million polluted sites worldwide and increasing due to the expansion of industrial, mining, smelting, and agricultural activities [1,2]. Metal mining industry, in particular, contributes significantly because inadequate management of mining waste (e.g., tailings and acidic waters) can release PHEs (As, Cd, Cu, Fe, Pb, and Zn) into the environment impacting the soil-plant-water system [3]. Therefore, proper management of these sites is essential to avoid damage to the environment and human health.

The use of Technosols specifically designed for the environmental problem in question is one of the most widely applied emerging ecotechnologies to remediate polluted areas [4]. Most Technosols applied in soil remediation are waste-based, as this ecotechnology also aims to deal with non-sustainable waste generation and contribute to the circular economy [5]. Besides general soil functions, Technosols have the capacity to promote different biogeochemical and edaphic processes (e.g., neutralise acidity, decrease sulphide oxidation, immobilise PHEs, increase soil fertility and biological activity, and reduce phytotoxicity), leading to the sustainability of the rehabilitation process in the medium and long term [6,7]. The effectiveness of this ecotechnology has been tested for different Technosols and climatic regions both in microcosm studies under controlled conditions [8,9,10,11] and in large-scale interventions [12,13,14,15].

Most studies on soil remediation evaluate the effectiveness of the applied ecotechnology by focusing on soil properties and PHE mobility/bioavailability [16,17], but additional assessments such as microbial community characterisation [18], ecotoxicity [19], and phytotoxicity tests [20] provide a more comprehensive understanding of the remediation outcomes.

Exposure to PHEs can cause various phytotoxic effects, such as growth inhibition, decreased photosynthesis and respiration, oxidative stress, altered enzyme activity and stomatal function, and DNA damage [21]. Plants can uptake PHEs through roots and subsequently be stored there or translocated to aerial parts, depending on the PHE and plant species [22]. Furthermore, this toxicity may differ depending on the interactions between PHEs, through antagonistic or synergistic functions in their uptake and translocation using, e.g., the same transporter [23,24].

PHEs can be taken up as free ions and soluble metal complexes, although most of them are captured, transported to the xylem, and translocated to shoots as chelates. It has been documented that most PHEs can be sequestered in roots as an adaptative response, causing no photosynthetic damage, due to various processes such as blockage by Casparian strips, chelation, compartmentalisation in rhizodermal vacuoles, and immobilisation in root cell walls [25,26]. However, when PHEs accumulate in sensitive parts, they can disrupt photosynthesis, e.g., by reducing the presence of photosynthetic pigments such as chlorophyll due to Fe and Mg deficiencies caused by PHEs [27] or by decreasing the efficiency of photosystem II through PHE interference with the electron transport chain [28].

The presence of PHEs also triggers the generation of reactive oxygen species (ROS) due to an imbalance in electron flow during photosynthesis or through Haber-Weiss reactions, leading to oxidative stress [29], for example, causing membrane destabilisation due to lipid peroxidation and subsequent formation of malondialdehyde (MDA) [29,30], increasing permeability for certain PHEs. In addition, elevated ROS levels in cells can lead to protein carbonylation and DNA damage [31,32]. To compensate for the effects of ROS, plants develop mechanisms to maintain cellular redox homeostasis, such as the induction of antioxidant systems based both on enzyme and non-enzymatic antioxidants [33,34,35], for example, the ascorbate-glutathione cycle [30,36] involved in the removal of H_2_O_2_ and lipid peroxides and the expression of stress-responsive genes [29]. Other more basic mechanisms rely solely on the synthesis of a metabolite such as proline (Pro) [34].

This study aims to evaluate the physiological response of lettuce (*Lactuca sativa* L.) plants grown on three Technosols employed in the remediation of PHE-polluted soils. This plant was selected because of its global consumption and its high sensitivity to pollution; in fact, it is among the species recommended to assess ecological effects of toxic substances [37,38]. Therefore, by assessing the reduction in phytotoxicity in lettuce through the measurement of plant growth, photosynthetic capacity, oxidative stress, and antioxidant defence, together with an evaluation of the capacity of Technosols to improve soil properties and reduce the solubility of PHEs, the effectiveness of this remediation ecotechnology can be proven.

## 2. Materials and Methods

### 2.1. Experimental Set-Up and Sampling

This study was conducted in the polluted areas remaining in the Guadiamar Green Corridor (Seville, Spain) more than 25 years after the Aznalcóllar mining accident (37°00′–37°30′ N, 6°10′–6°20′ W). The tailings pond breach at the Aznalcóllar pyrite mine (April 1998) resulted in the discharge of 3.6 hm^3^ of acidic water and 0.9 hm^3^ of toxic tailings with high concentrations of PHEs (i.e., Zn, Cu, Cd, As, and Pb), causing severe soil and water pollution. Several remediation actions were implemented, namely removal of potentially toxic tailings and highly polluted topsoils, followed by assisted natural remediation based on the application of organic and inorganic amendments and on phytostabilisation with native vegetation [39], achieving remediation of almost the entire affected area. However, some areas remain residually polluted, posing a long-term toxic risk to living organisms in the area [16,40,41]. For this work, a composite sample was taken from these soils polluted by PHEs (i.e., As, Cd, Cu, Fe, Pb, and Zn) (PS) at 0–10 cm depths at five different locations identified by the absence of vegetation using satellite images [16]. Furthermore, a composite sample of recovered soil (RS), consisting of soils affected by the spill but now recovered due to the applied remediation actions, was taken in the vicinity of the polluted patches at 0–10 cm depths to serve as a control (Appendix A).

The assay consisted of five treatments placed in pots of 1.6 dm^3^ volume with eight pots per treatment. The treatments were (i) recovered soil (RS), (ii) polluted soil (PS), (iii) Technosol TO, (iv) Technosol TS, and (v) Technosol TV. Technosols were designed and produced by mixing 60% of polluted soil (PS) with 18% of organic-rich waste (TO: composted solid olive-mill by-product [OL], TS: composted sewage sludge [WS], and TV: vermicompost from pruning and gardening waste [VC]), 20% of marble cutting and polishing sludge [MS], and 2% of iron oxyhydroxide-rich sludge [IO] (Figure 1). These amendments were selected to provide the Technosols with the capacity to cope with the polluted status in PS, with proportions based on the actual doses applied in the field in previous remediation actions in this affected area [42]. Marble sludge provides pH buffering capacity; organic wastes (OL, WS, and VC), rich in organic matter, support the recovery of biological activity and contribute to soil aggregation; and iron-rich waste (IO) promotes the retention capacity of anionic PHEs (i.e., As and Sb). Characterisation with the physicochemical properties and PHE contents of these wastes can be found in Appendix A extracted from [43]; furthermore, it should be noted that they are non-hazardous, and in particular, the organic ones are sanitised and pathogen-free due to the composting process and the raw material used [44,45]. To produce the Technosols, wastes were dried, sieved to <4 mm, and mixed manually together with the dry polluted soil (total fraction) in plastic trays. Technosols were incubated for two months at room temperature (20–25 °C) with regular watering and aeration (every three days) to maintain 70% of the field capacity.

One seedling of *Lactuca sativa* (L.) cv. Phillipus per pot and treatment was planted. These seedlings were obtained from seed germination and growth for 35 days in cell flats (90 cm^3^) filled with perlite placed on benches in an experimental greenhouse in southern Spain (Granada, Motril, Saliplant S.L.). After transplanting, the seedlings were transferred to a growth chamber under controlled environmental conditions: temperature 25/15 °C and photoperiod 16/8 h (day/night), relative humidity 60–80%, and 350 μmol m^−2^ s^−1^ photosynthetically active radiation (PAR) measured with a 190 SB quantum sensor (LI-COR Inc., Lincoln, NE, USA). The plants were watered by capillary irrigation in lightweight polypropylene trays with 3 l volume. Seven days after transplanting, acute phytotoxic effects (i.e., chlorosis, necrosis, and reduced plant growth) started to be observed in some treatments (PS and TS), so ten days later, all *L. sativa* plants from each treatment were collected. Full expanded leaves and roots were washed with distilled water, dried on filter paper, and weighed to obtain fresh weight (FW). Half of the roots and leaves of each treatment were frozen at −40 °C for biochemical characterisation, and the other half of the plant materials was lyophilised to obtain dry weight (DW) and to analyse the PHE concentrations. The experimental design consisted of a randomised complete block with five treatments (RS, PS, TO, TS, and TV), arranged in individual benches with eight plants per treatment and three replications each.

### 2.2. Soil Analysis

A composite sample per treatment was taken from 100 g aliquots of soil from each pot. From this soil sample, three replicates per treatment (RS, PS, TO, TS, and TV) were taken and subsequently air-dried at room temperature, homogenised, and sieved at 2 mm, and a small fraction was finely ground for the following characterisation: pH in water (1:2.5 m:V) with 914 pH/conductometer (Metrohm AG, Herisau, Switzerland); electrical conductivity (EC) in water (1:5 m:V) using an Eutech CON700 conductometer (Oakton Instruments, Vernon-Hills, IL, USA); organic carbon (OC) content by wet oxidation [46]; calcium carbonate (CaCO_3_) content by volumetric gases using a modified Bernard calcimeter [47]; cation exchange capacity (CEC) and extractable Na^+^ by the ammonium/sodium acetate method [48]; and total and water-soluble PHE concentrations (As, Cd, Cu, Fe, Pb, and Zn). Total PHE concentrations were extracted by acid digestion in aqua regia (HNO_3_ + HCl 3:1 V:V) in a Mars XP1500 Plus microwave (CEM Corporation, Matthews, CN, USA) and soluble PHE concentrations were determined with distilled water (1:5 m:V) [49]. Both fractions were measured by inductively coupled plasma mass spectrometry (ICP-MS) using a PerkinElmer NexION 300D spectrometer (PerkinElmer, Inc., Waltham, MA, USA). The precision and accuracy of this method were evaluated by testing (*n* = 3) the certified reference material CRM 052–050 (RT-Corporation Limited, Salisbury, UK) and using procedural blanks. Calibration standards were also prepared from ICP single-element standard solutions (Merck, Darmstadt, Germany) after dilution with 10% HNO_3_. The measured values for all elements were within the prediction range of the certified value (Appendix A).

### 2.3. Plant Analysis

#### 2.3.1. Relative Growth Rate (RGR) Determination

To determine the RGR, before starting the assay (*T_i_*), the leaves of eight *L. sativa* seedlings from the Saliplant S.L. nursery were lyophilised to record dry weight (DW). After 17 days of growth (*T_f_*), DW was also determined. Thus, the RGR can be calculated from the increase in leaf DW from the beginning to the end of cultivation using the following equation: *RGR* = (*ln DW_f_* − *ln DW_i_*)/(*T*), where *T* is the assay duration (17 days).

#### 2.3.2. PHE Concentrations in Plants

The composite sample of lyophilised plants, separated into aerial part and roots, was ground, and then, three replicates per treatment were taken. Acid digestion in aqua regia (HNO_3_ + HCl 3:1 V:V) was performed in a Mars XP1500 Plus microwave (CEM Corporation, Matthews, CN, USA) to measure PHE concentrations (As, Cd, Cu, Fe, Pb, and Zn) in their tissues by ICP-MS in a PerkinElmer NexION 300D spectrometer (PerkinElmer, Inc., Waltham, MA, USA). The precision and accuracy of this method were assessed by measuring (*n* = 3) the certified reference material of lichen BCR^®^-482 (EC-JRC-IRMM, Geel, Belgium). The analysis of the CRM gave satisfactory results for all elements considered (Appendix A).

#### 2.3.3. Leaf Gas Exchange Parameters

Leaf gas exchange parameters were recorded with an infrared gas analyser LICOR 6800 Portable Photosynthesis System (IRGA: LI-COR Inc., Lincoln, NE, USA). Intermediate leaves of three plants for each treatment were placed in a leaf cuvette set with optimal growth conditions. The instrument was warmed up at 25 °C for 30 min and calibrated before measurements, with standard optimum cuvette conditions of 400 μmol mol^−1^ CO_2_ concentration, 500 μmol m^2^ s^−1^ PAR, 70% relative humidity, and room temperature. Net photosynthesis rate (ACO_2_), transpiration rate (E), stomatal conductance (g_s_), and intercellular CO_2_ (C_i_) were recorded. Data were analysed with Photosyn Assistant software v. 3 (Dundee Scientific, Dundee, Scotland, UK). Instantaneous water use efficiency (WUE) was calculated as the A/E quotient.

#### 2.3.4. Oxidative Stress Indicators and Antioxidant Activity

Malondialdehyde (MDA) concentration was determined according to the method of Fu and Huang [50] by measuring absorbance at 532 nm. The non-specific absorbance value at 600 nm was obtained to correct for turbidity. MDA concentration was calculated using 155 mM^−1^ cm^−1^ as an extinction coefficient. Proline (Pro) concentration was measured using the method adapted from Bieleski and Turner [51] by measuring absorbance at 515 nm and using standard curves made from Pro stock solutions.

Two reactive oxygen species (ROS) were also determined: hydrogen peroxide (H_2_O_2_) and superoxide ion (O_2_^−^). The H_2_O_2_ content of the leaf samples was determined by colorimetry according to the method of Junglee et al. [52], based on a reaction with potassium iodide and an absorbance reading at 350 nm compared to H_2_O_2_ standard curves. For O_2_^−^ content, the method of Kubiś [53] was followed, where the ability of extracts to reduce nitroblue tetrazolium (NBT) was determined by measuring absorbance at 580 nm and using standard curves from sodium nitrite.

Superoxide dismutase (SOD; EC 1.15.1.1) activity was determined by inhibition of the photochemical reduction of NBT according to the methods of Giannopolitis-Ries and Beyer-Fridovich modified by Yu [54,55,56]. One unit of SOD activity was defined as the amount of enzyme required to cause 50% inhibition of the reduction of NBT as monitored at 560 nm. Catalase (CAT; EC 1.11.1.6) activity was determined by the methods of Nakano-Asada and Rao et al. [57,58], based on the measurement of H_2_O_2_ consumption at 240 nm. The protein concentration of the extracts was determined according to Bradford [59], using bovine serum albumin as a standard.

#### 2.3.5. Ascorbate-Glutathione Cycle

The concentrations of total glutathione (GSH Total) and oxidised glutathione (GSH Oxid) were determined according to Noctor and Foyer’s method [60]. Reduced glutathione (GSH Red) content was estimated from the difference between total and oxidised glutathione. The concentrations of total ascorbate (AsA Total) and its reduced form (AsA Red) were determined by the method of Law [61]. Standard curves made from stock solutions of each of these compounds (GSH Total, GSH Oxid, AsA Total, and AsA Red) were used to measure their concentration.

The method described by Rao [62] was used to determine both ascorbate peroxidase (APX; EC 1.11.1.11) and glutathione reductase (GR; EC 1.6.4.1). For APX activity, the disappearance of H_2_O_2_ was measured at 290 nm, and for GR activity, the oxidation of NADPH was measured at 340 nm. The protein concentration of the extracts was determined according to Bradford [59], using bovine serum albumin as a standard.

All biochemical parameters analysed in this and the previous section were performed with an Infinite 200 Nanoquat spectrophotometer (Tecan, Männedorf, Switzerland).

### 2.4. Statistical Analysis

Data were subjected to a simple ANOVA at 95% confidence using Statgraphics Centurion 16.1.03 software. Differences between treatment means were compared using Fisher’s least-significant difference (LSD) test. Significance levels are expressed as * *p* < 0.05, ** *p* < 0.01, *** *p* < 0.001, or NS (not significant).

## 3. Results and Discussion

### 3.1. Soil Properties and PHE Levels

The polluted soil (PS) was characterised by an extremely acid pH, as opposed to the neutral to slightly alkaline pH of recovered soil (RS) (Table 1), due to the formation of sulphuric acid by oxidation of sulphides contained in the pyritic sludge dumped in the Aznalcóllar mining accident [63]. For this reason, one of the remediation actions undertaken immediately afterwards was the addition of calcium carbonate-rich amendments to neutralise acidity and consequent mobilisation of PHEs due to their high pH buffering capacity [64]. This explains the low CaCO_3_ content in RS (Table 1), as partial dissolution of carbonates occurred to buffer pH [65]. In this regard, because pH exerts a strong influence on the mobility of PHEs, which generally decreases with increasing pH [40], the Technosols have been designed with a composition of 20% carbonate-rich waste, the marble sludge (MS), to provide them buffering capacity. Consequently, the Technosols TO, TS, and TV, despite the high percentage of PS in their composition, had neutral to slightly alkaline pH and considerable CaCO_3_ content (Table 1).

Likewise, PS showed low organic carbon (OC) content and low cation exchange capacity (CEC) (Table 1), which limits both chemical and physical fertility. These soils are not only poor in nutrients but also tend to crust, which further limits the possibility of vegetation development [66]. In contrast to RS, where previous remediation actions allowed the growth and development of plants, significantly increasing the soil organic matter compared to PS [67]. For this reason, an 18% organic amendment was included in the three Technosols, resulting in a much higher OC content than PS. However, there was a large variability in OC content among Technosols (TO: 4.1%, TS: 5.6%, and TV: 2.1%) due to the different OC richness of the organic wastes (Appendix A). In addition, the nature of the organic waste can lead to variations in the evolution of the organic matter during the incubation period along the Technosol production, thereby contributing to differences in the fertility status of each Technosol [11]. In fact, TO had a higher CEC than TS despite its lower OC content. Thus, the composted solid olive-mill by-product (OL) used in TO was much more reactive than the sewage sludge (WS) used in TS. In the case of TV, the decrease in its CEC compared to the others was due to the lower OC input with the vermicompost from pruning and gardening waste (VC), which presumably is also more easily mineralised (Table 1 and Appendix A).

The high electrical conductivity (EC) values in PS (Table 1) were a consequence of the oxidation of sulphides to soluble sulphates in the pyritic sludge upon drying [63]. Technosols include 60% PS, which justifies the EC levels in them (Table 1). However, TO and TS showed a considerably higher EC than PS and TV because of the organic wastes used (Appendix A). The same occurred with exchangeable sodium (Na^+^), which showed very high values in TO and TS (Table 1), even reaching levels that can cause stress for plant development and compromise their growth [68].

At present, despite the remediation measures implemented by the Regional Government of Andalusia and more than 25 years after the accident, some PHEs (As, Cu, Pb, and Zn) are still of concern in the area because of their potential environmental risk due to their solubility and long-term bioavailability [18], especially in areas close to the mine without vegetation. Indeed, total soil concentrations of As and Pb in PS (mg kg^−1^; As: 346, Pb: 546; Table 2) exceeded the regulatory levels of 36 and 275 mg kg^−1^ [69] by 9.6 and 2, respectively. In any case, these remediation efforts were effective in almost the entire affected area [39,42], leading not only to an improvement in soil conditions but also to decreased PHE concentrations. As shown in our results, the total concentrations of most PHEs in RS were significantly lower than in PS (Table 2). Except for Zn, whose total concentration in RS was higher than in PS because its solubility is strongly controlled by pH. In PS, Zn has leached intensively over time due to soil acidity, thus reducing its total concentration compared to RS [70,71].

The potential toxicological risk of these PHEs depends largely on their mobility, rather than their total concentrations [72,73], since the water-soluble fraction is the most readily accessible to plants and living soil organisms and can therefore cause damage to the ecosystem and/or enter the trophic chain [74,75]. Therefore, it is essential to assess their mobility, as the risk of pollution could be over- or under-estimated if it is only based on total concentrations. Furthermore, the remediation of residually polluted soils with Technosols is not expected to modify the total PHEs content but will reduce their mobility and phytoavailability. Indeed, the total PHE concentrations in Technosols were high, as 60% of them were composed of PS, although in TS, the concentration of Zn_T_ was even higher because of Zn concentration in sewage sludge (Appendix A).

The solubility of PHEs in Technosols changed strongly compared with that of PS. The high water-soluble concentrations of Cd, Cu, and Zn in PS decreased significantly in most Technosols to values roughly close to those in RS (Table 2). Because the solubility of Cd, Cu, and Zn is strongly conditioned by soil properties, it decreases as pH rises [76]. The pH neutralisation in Technosols was mediated by carbonates [77] provided by the marble sludge. Furthermore, the increased OC content in Technosols may also contribute to the decrease in their solubility [76,78]. However, in TO, Cu_W_ concentration did not decrease with respect to PS, and in contrast, it was even higher (Table 2); this may be due to the fact that Cu is not as pH-dependent as Cd and Zn [79] and that this element has a strong affinity for organic matter [18,80]. This was in agreement with our data, as the organic waste used in TO (solid olive-mill by-product: OL) was characterised by being more reactive than the other two organic wastes (WS and VC). This organic waste (OL) showed the highest CEC (Appendix A), which could allow for the adsorption of a greater amount of Cu in exchangeable form that can pass into the soil solution and, thus, be more available to the plant. Furthermore, soluble Cd and Zn presented significantly lower concentrations in TV than in TO and TS (Table 2), with the only difference between Technosols being the organic waste added. On the other hand, although the same proportion of iron oxyhydroxide-rich sludge was used in all Technosols, Fe_W_ concentration was significantly higher in TS (Table 2) due to the contribution of more forms in this Technosol from the added organic waste, sewage sludge (WS) (Appendix A) [43]. Hence, the nature and composition of the components of the Technosols are decisive in characterising the solubility of PHEs.

Water-soluble As in Technosols increased significantly compared to PS, varying among Technosols (Table 2). Arsenic solubility is strongly conditioned by pH; however, the acidity of the soil solution decreases the presence of soluble forms of As, whereas they increase at pH close to neutrality [81]. In this sense, Simón et al. [65] pointed out that the liming of soils polluted by PHEs should be performed cautiously because above pH 6.5, this element can be solubilised. To counteract this, 2% of iron oxyhydroxide-rich sludge was added to all Technosols for the immobilisation of As, since in soils, it usually appears in anionic forms (HAsO_4_^2−^, H_2_AsO_4_^−^), so it can be fixed to the positive charges that dominate in iron oxyhydroxides [81,82]. The significant increase in As_W_ concentration in TO and TS compared to PS (Table 2) may be due to the potential competition of arsenates with soil organic matter for Fe oxyhydroxide binding sites [42,83,84]. Arsenic solubility in TV, although higher than in PS, was significantly lower compared to TS and TO, due to the lower OC content in the former decreasing the competitive effect of organic matter for As fixation sites.

A similar behaviour was observed with Pb solubility, where the higher organic matter content in RS, TO, and TS could lead to an increase in Pb_w_. On the contrary, TV did not show an increase in Pb solubility compared to PS, with the OC content being the lowest among the three Technosols (Table 1 and Table 2). The relationship between Pb solubility and organic matter content is in agreement with the results of several authors [42,76], who observed a slight increase in Pb solubility in reclaimed soils of the Guadiamar Green Corridor compared to soils that remain polluted and have very low OC content. However, several authors have highlighted the affinity of Pb for organic matter [83,85,86], which can interact with it in a nearly stable way, which can condition its solubility. Therefore, the controversy in this respect highlights the need for further research on the interaction between organic matter and Pb in polluted soils.

### 3.2. Growth of L. sativa

Figure 2 shows that the worst physiological state of *L. sativa* plants was in PS and TS, with smaller size, fewer leaves, and a more necrotic appearance. Moreover, Technosol TV showed similar phenotypic characteristics to the plants grown on RS. This growth dynamic could be due to the fact that stress factors such as soil acidity; salinity; and finally, exposure to certain PHEs caused a decrease in plant growth [25,34].

Indeed, the highest relative growth rates (RGR) were recorded for *L. sativa* plants grown on RS and TV, whereas the lowest were on PS and TS (Figure 3). According to these results, the lower growth of *L. sativa* plants may be due to the higher toxicity resulting from the presence of PHEs (Table 2), as has been observed in several studies. Navarro-León et al. [87] reported that exposure to Cd toxicity caused a 60% decrease in the leaf biomass of *Brassica rapa* L. subsp. *trilocularis* (Roxb.). Similarly, Zaier et al. [88] showed that the presence of Pb strongly inhibited the growth of *Brassica juncea* (L.).

### 3.3. PHE Accumulation in L. sativa

High concentrations of PHEs present in polluted areas can exert a detrimental effect on plants, which respond at the molecular, cellular, physiological, anatomical, and morphological levels [89]. Thus, plants have evolved, over time, complex mechanisms of hyperaccumulation, tolerance, exclusion, and chelation with organic molecules to overcome this abiotic stress [90]. In this case, translocation processes seem not to be prominent because the accumulation of PHEs in the aerial part of *L. sativa* plants was much lower than in roots for all treatments and for all PHEs (Table 3). This could be a result of the Casparian strip, which acts like a selective barrier by blocking the entry of a large part of PHEs into the xylem to ensure selective ion transport while retaining ions [91,92,93].

In treatments RS and TV, the highest concentrations of Cd and As were found in the aerial part. However, no accumulation was observed for Pb and Cu in the same treatments (Table 3). This is probably due to the low solubility of both elements in TV and Pb in RS (Table 2), as well as the competition between certain PHEs for transporters and possible antagonistic effects [94]. The latter could be the case of Cd and Zn, which although both were at high concentrations in TV, their combined effect could result in less stress than either of them separately [95]. On the other hand, at high concentrations, a potent synergy between the two elements has also been shown [94]. In this study, TS showed the highest concentrations of PHEs such as Pb, Fe, and Cu in the shoots of *L. sativa* plants, even higher than those grown in PS (Table 3). However, the higher solubility of pH-dependent elements such as Cd, Cu, and Zn in PS resulted in higher root concentrations than in the other treatments. In contrast, As and Pb, generally less mobile at acidic pH, showed the highest root concentrations in the more organic soils at neutral to slightly alkaline pH (RS, TO, TS, and TV), especially in Technosols with two- to three-fold higher concentrations than in RS (Table 3). Therefore, toxicity may not be directly related to higher PHE concentrations due to their compartmentalisation and chelation by binding to different ligands for accumulation in shoots and roots in a non-toxic way [25], as seems to be occurring in TV, where, despite the lower solubility for Cd, Cu, and Zn, high concentrations of these PHEs were accumulated in the *L. sativa* roots.

### 3.4. Photosynthesis and Gas Exchange in L. sativa

A reduction in the process of photosynthesis and gas exchange due to alterations in the photosynthetic apparatus caused by the presence of PHEs impairs the development of the plant, resulting in various damages, such as reduced growth and nutrient uptake [25,27]. In this sense, *L. sativa* plants grown on PS and TS showed a drastic depletion in photosynthetic and transpiration rates, as well as stomatal conductance compared to RS (Table 4), possibly due to the damage caused by PHEs, as well as other stress conditions such as soil acidity (Table 1) [28,96,97]. Moreover, the low values of intracellular CO_2_ concentration in plants grown in PS and TS (Table 4) may indicate that not only was the photosynthetic apparatus damaged but also that stomatal conductance limitations could have caused a decrease in rubisco activity and thus the collapse of photosynthesis [98]. The same tendency was observed in plants grown on TO for all parameters except intracellular CO_2_ concentration, which was similar to RS (Table 4). Therefore, these low photosynthetic rates in Technosols TO and TS, and especially in PS, besides the RGR values (Figure 3), could be due to the high PHE toxicity problem in the soil of these treatments [28]. Moreover, the severe reduction in transpiration, resulting from stomatal closure, led to an increase in water use efficiency (WUE) in both TS and PS. This improvement is likely due to an increase in soil solute potential caused by higher contents of soluble PHEs in the soil solution. The effect may be more pronounced in TS, with increased quantities of Na^+^ ions (Table 1) that further raises the soil solute potential, intensifying the water-saving response [68]. On the other hand, transpiration drives the transport and translocation of ions to the leaves; therefore, treatments in which plants showed a higher transpiration rate together with a better physiological state may accumulate more PHEs in the aerial part [99], as the data showed in treatments RS and TV (Table 3 and Table 4). Finally, the values for all these growth and photosynthetic parameters for TV were similar to those for RS (Figure 3, Table 4), showing the great effectiveness of this Technosol in the remediation of polluted soils, allowing the establishment of healthy vegetation. As in previous studies [11], it was in this same Technosol where *Trifolium campestre* Schreb. grew with the highest biomass. Similarly, other Technosols amended with green waste compost were shown to be very effective in urban greening, as compost improves soil nutrient content and meets plant needs, leading to higher photosynthetic efficiency and carbon investment in the herbaceous species *Malva sylvestris* L. (higher leaf area and lower leaf mass per area) early on and higher leaf water content in the Mediterranean sclerophyll species *Phillyrea angustifolia* L. and *Quercus ilex* L. in the long term [100].

### 3.5. Oxidative Stress and Antioxidant Activity in L. sativa

In *L. sativa* plants grown on PS, the malondialdehyde (MDA) generated was twelve-fold higher than in plants grown on RS, while this increase was only five-fold greater in the other treatments compared to RS (Figure 4a). This higher MDA concentration produced in lipid peroxidation responds to oxidative stress caused by PHE toxicity and other negative conditions such as salinity [29,30]. These results were consistent with other previous reports [30,101], where an increase in the MDA levels of 126% and 30% were reported in *L. sativa* plants exposed to toxic concentrations of Zn and Fe. Likewise, proline (Pro) levels also increased in PS and TS compared to RS, although the highest values were recorded in TS (Figure 4b). Similar results were observed by Yang et al. [102] when assessing Pb toxicity in *Tritium aestivum* L., with a 207% increase in Pro concentration under these unfavourable conditions. Thus, the increased synthesis of this amino acid can be interpreted as an adaptive mechanism under the stress conditions of PS and TS, as Pro is involved in reactive oxygen species (ROS) extinction processes by functioning as an antioxidant metabolite and having the capacity to chelate PHEs [29,34]. This stress was more pronounced in PS than in TS, as reflected by the lower MDA concentration recorded (Figure 4a). This could be precisely due to the higher proline synthesis reported in TS, as it also has an osmoprotective behaviour and is a target of the hydroxyl radical, which could mitigate the effects of salinity and oxidative stress, as higher Fe and Cu concentrations in the aerial part of plants grown in TS (Table 3) could be responsible for the formation of the hydroxyl radical through Haber-Weiss and Fenton reactions [34,68]. It probably prevents ROS concentrations in TS by ultimately leading to lipid peroxidation and MDA formation (Figure 4a). In fact, Siripornadulsil et al. [103] correlated a higher Pro content with a mitigation of oxidative damage (lower MDA) in the presence of PHEs.

Regarding ROS production, significant differences of O_2_^−^ concentration were observed in PS and TS with respect to RS (Figure 5a), as well as H_2_O_2_ production, which was also higher in TO (Figure 5b). The increased production of ROS is a very common effect in plants subjected to abiotic stresses such as PHE toxicity, known as oxidative burst [29,34]. Consistent with our results, some studies [30,101] showed that plant exposure to toxic concentrations of Fe and Zn lead to an increased ROS production in *Brassica oleracea* L. and *L. sativa*. Also, antioxidant enzymatic activity was measured, and the data showed that superoxide dismutase (SOD) and catalase (CAT) activities were increased in PS, TO, and TS with respect to RS. As in the other assays, the highest values were observed in PS and TS (Figure 5c,d). In this study, the greater ROS production in Technosols TO and TS, and especially in PS, was combined with an increase in antioxidant enzymatic activity that was not observed in either RS or TV (Figure 5c,d). This higher antioxidant enzymatic activity occurred to mitigate plant oxidative stress, such as SOD activity, which transforms O_2_^−^ into less reactive H_2_O_2_, and CAT activity, which removes H_2_O_2_ [29,102]. However, O_2_^−^ and H_2_O_2_ remained high in TS and PS, despite high SOD and CAT activities (Figure 5), thus reflecting a higher degree of stress. Furthermore, the H_2_O_2_ concentration at TS was decreased not only by CAT activity but also by high Fe and Cu contents in the aerial part (Table 3), which resulted in Haber-Weiss and Fenton reactions to form the much more reactive hydroxyl radical [29]. On the other hand, the high H_2_O_2_ concentration in TO could be related to lower CAT activity but high SOD activity, which transformed much of O_2_^−^ to a concentration similar to RS (Figure 5). Finally, it is worth noting again the similarity between TV and RS, which would reveal a low stress level due to the significant efficacy of this Technosol.

### 3.6. Ascorbate-Glutathione Metabolism in L. sativa

The ascorbate-glutathione cycle is another important mechanism for plants to limit oxidative stress, maintain cell redox status, and decrease ROS levels. This network involves important antioxidant compounds and enzymes with great antioxidant capacity and a special protection function against oxidative damage caused by exposure to PHEs [29]. In this study, the highest total glutathione (GSH Total) concentrations were recorded in PS, TS, and TV compared to RS, with the highest values in PS and TS. As for oxidised glutathione (GSH Oxid), an increase in this compound was observed in PS compared to RS (Figure 6a). Likewise, a large decrease in GSH Oxid in relation to GSH Total was observed only in TS (Figure 6a), revealing that more reduced glutathione (GSH Red) could be available in TS. On the other hand, compared to RS, plants grown in PS registered the highest concentration of total ascorbate (AsA Total), followed by those grown in TS, and lastly, those grown in the other Technosols with similar values (Figure 6b). However, the reduced ascorbate form (AsA Red) increased significatively in PS (Figure 6b). Glutathione reductase (GR) activity was increased in PS, TS, and TV compared to RS (Figure 6c). Finally, the ascorbate peroxidase (APX) activity was higher for PS and all Technosols, and the highest values were found in PS and TS, followed by TO and TV with intermediate values, and RS with the lowest values (Figure 6d). Our results were similar to other studies that reported an increase in GSH Total concentration, and especially in GSH Red, to mitigate the oxidative stress caused by Pb and Zn toxicity in *T. aestivum* and *B. oleracea* plants [30,104]. Moreover, the lower glutathione reductase (GR) activity in PS with respect to TS (Figure 6c) revealed a higher stress condition in PS, as Barrameda-Medina et al. [30] described for *L. sativa* plants exposed to high Zn concentrations. This fact would explain why our results showed a higher GSH Oxid accumulation and less glutathione recycling in PS than in TS. Furthermore, the availability of GSH not only contributes to the reduction in ROS but also contributes to the chelation of PHEs, either directly through its reduced form or by the formation of phytochelatins [105]. Ascorbate is another antioxidant non-enzymatic metabolite with a function similar to that of glutathione in ROS detoxification. In this sense, the existence of a large difference between AsA Total concentration and AsA Red in TS and especially in PS (Figure 6b) showed that this metabolite was being oxidised to mitigate the abiotic stress by PHEs [30,101], which did not occur in RS and TV, possibly due to the lower stress conditions and lower APX activity with respect to PS and TS (Figure 6d), consistent with Buturi et al. [101]. Moreover, the higher APX activity achieved at TS also accounted for the lower H_2_O_2_ concentration compared TO. Finally, the degree of stress generally leads to an increase in the activity of antioxidant enzymes such as SOD, CAT, APX, and GR (Figure 5c,d and Figure 6c,d), aimed at reducing oxidative stress caused by PHEs, as occurred in PS [29,104].

## 4. Conclusions

The three designed Technosols (TO, TS, TV), composed of polluted soil (PS) with a mixture of organic and inorganic wastes from local industries (mines, urban gardening services, wastewater treatment services, and olive mills), improved the unfavourable conditions of PS (e.g., neutralised acidity and increased OC content) and, consequently, modified the mobility of most PHEs. However, Technosols induced a differential response in each case, mainly influenced by the nature of the organic waste. Thus, the solubility of Cd, Cu, and Zn decreased in all Technosols (except Cu in TO, which increased), while As was re-solubilised in all Technosols and Pb solely in TO and TS. Moreover, the added wastes not only affect to PHEs solubility but may even represent an input of PHEs. This is the case of TS with respect to water-soluble Fe and total Zn, or TO and TS with Na^+^ ions contributing to the exchange complex causing salt stress.

The presence of PHEs, especially in their soluble form, caused oxidative stress in *L. sativa* plants grown in TS and especially in PS, triggering a strong antioxidant response as a tolerance mechanism. The detoxification of ROS took place through the production of antioxidant metabolites such as glutathione and ascorbate, as well as increased enzymatic activity (i.e., catalase, superoxide dismutase, ascorbate peroxidase, and glutathione reductase) to mitigate their damage effect. Another effect of PHE exposure was the limitation of photosynthesis and gas exchange processes; in both TS and PS, the photosynthetic rate was reduced, leading to a decrease in *L. sativa* plant growth due to the stomatal limitations and damage to the photosynthetic machinery caused by PHEs.

Thus, the ecotechnology of Technosols can constitute a potential solution for the remediation of persistent polluted soils, although, in particular, the Technosol composed of pruning and gardening vermicompost (TV) showed an overall performance far superior to that of the other Technosols. In addition to improving soil properties and reducing the mobility of most PHEs, phytotoxicity evaluated in *L. sativa* plants grown on TV was minimal, with great similarity to the recovered soil in most of the parameters analysed. However, there are some potential risks that need to be further investigated and taken into account, such as the potential increase in solubility of As and Pb in Technosols due to increased OC and the input of potentially toxic substances (i.e., Fe, Zn, and Na ions) from waste materials used as amendments for Technosol construction.

## Figures and Tables

**Figure 1 plants-13-03222-f001:**
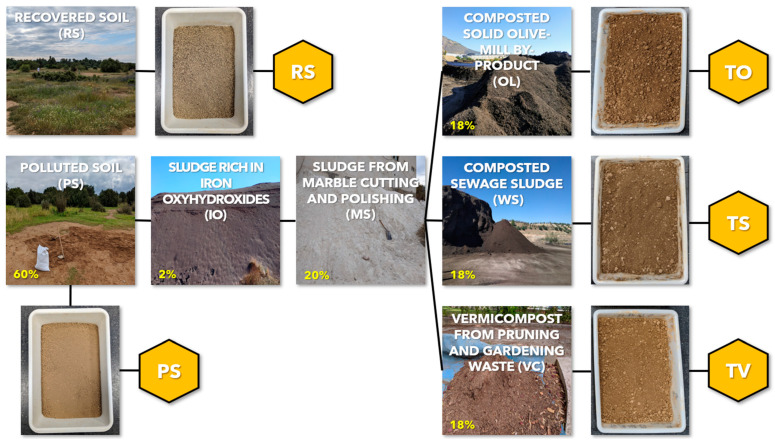
Scheme of the treatments: control soils (polluted soil [PS] and recovered soil [RS]) and constructed Technosols (TO, TS, and TV), including their compositions (TO: 60% PS + 2% IO + 20% MS + 18% OL; TS: 60% PS + 2% IO + 20% MS + 18% WS; and TV: 60% PS + 2% IO + 20% MS + 18% VC).

**Figure 2 plants-13-03222-f002:**
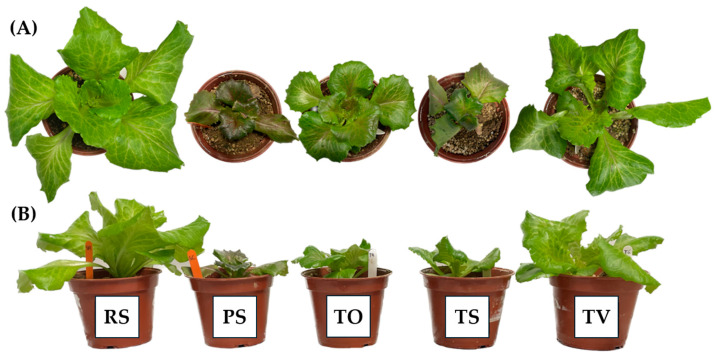
Physiological state of *L. sativa* plants grown in recovered soil (RS), polluted soil (PS), and Technosols (TO, TS, and TV) after 17 days. View from top (**A**) and side (**B**).

**Figure 3 plants-13-03222-f003:**
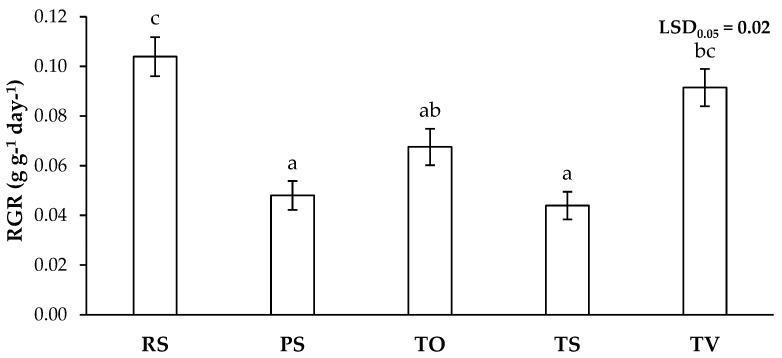
Relative growth rate (RGR) of the aerial part of *L. sativa* plants grown in recovered soil (RS), polluted soil (PS), and Technosols (TO, TS, and TV) (*n* = 9). Letters indicate statistically significant differences between treatments (LSD test, *p* < 0.05).

**Figure 4 plants-13-03222-f004:**
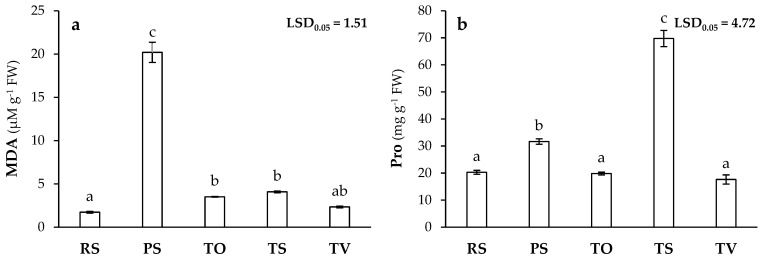
Malondialdehyde (MDA) (**a**) and proline (Pro) (**b**) concentrations in *L. sativa* plants grown in recovered soil (RS), polluted soil (PS), and Technosols (TO, TS, and TV) (*n* = 9). Letters indicate statistically significant differences between treatments (LSD test, *p* < 0.05).

**Figure 5 plants-13-03222-f005:**
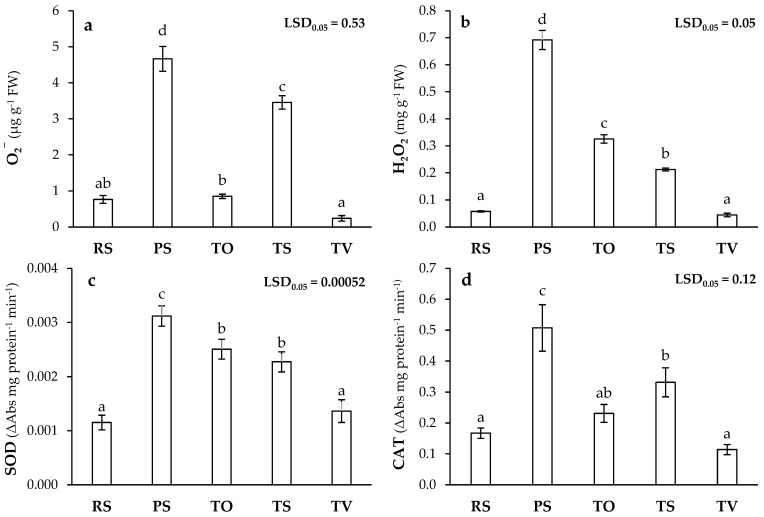
Reactive oxygen species (ROS) [(**a**): superoxide ion (O_2_^−^), (**b**): hydrogen peroxide (H_2_O_2_)] and associated enzymatic activities [(**c**): superoxide dismutase (SOD), (**d**): catalase (CAT)] in *L. sativa* plants grown in recovered soil (RS), polluted soil (PS), and Technosols (TO, TS, and TV) (*n* = 9). Letters indicate statistically significant differences between treatments (LSD test, *p* < 0.05).

**Figure 6 plants-13-03222-f006:**
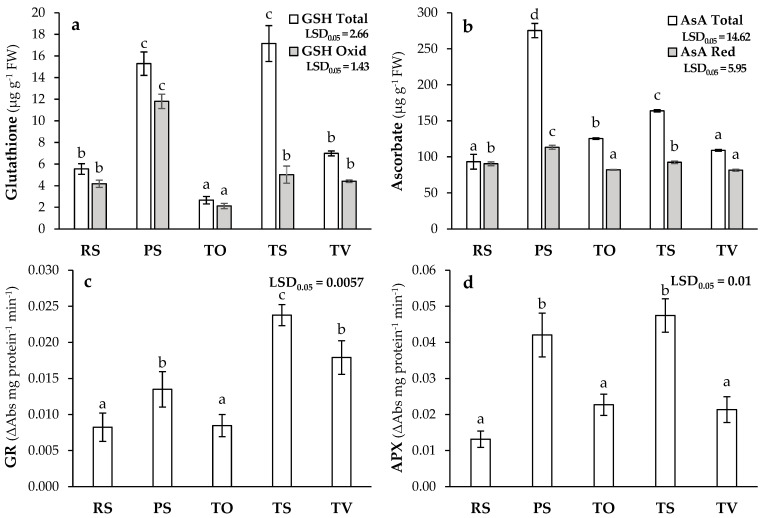
Antioxidant metabolites [(**a**): total glutathione (GSH Total) and oxidised glutathione (GSH Oxid), (**b**): total ascorbate (AsA Total) and reduced ascorbate (AsA Red)], and associated enzymatic activities [(**c**): glutathione reductase (GR), (**d**): ascorbate peroxidase (APX)] in *L. sativa* plants grown in recovered soil (RS), polluted soil (PS), and Technosols (TO, TS, and TV) (*n* = 9). Letters indicate statistically significant differences between treatments (LSD test, *p* < 0.05).

**Table 1 plants-13-03222-t001:** Main properties of recovered soil (RS), polluted soil (PS), and Technosols (TO, TS, and TV).

Soils	pH (H_2_O) _1:2.5_	EC _1:5_ (dS m^−1^)	OC (%)	CaCO_3_ (%)	CEC (cmol_+_ kg^−1^)	Na^+^ (cmol_+_ kg^−1^)
**RS**	7.38 ± 0.12 c	0.96 ± 0.11 a	5.89 ± 0.24 e	0.80 ± 0.09 a	35.28 ± 3.41 d	0.60 ± 0.02 a
**PS**	4.16 ± 0.09 a	2.99 ± 0.18 b	0.87 ± 0.09 a	0.56 ± 0.06 a	11.25 ± 1.76 a	0.53 ± 0.03 a
**TO**	7.72 ± 0.15 e	4.96 ±0.21 d	4.08 ± 0.14 c	23.06 ± 1.25 b	28.50 ± 2.64 c	2.45 ± 0.13 b
**TS**	7.30 ± 0.08 b	4.34 ± 0.17 c	5.59 ± 0.17 d	23.17 ± 1.21 b	24.34 ± 1.74 c	2.46 ± 0.16 b
**TV**	7.62 ± 0.13 d	2.96 ± 0.15 b	2.09 ± 0.11 b	22.83 ± 1.51 b	16.51 ± 1.66 b	0.70 ± 0.02 a
***p*-value**	***	***	***	***	***	***
**LSD_0.05_**	0.02	0.08	0.14	0.77	4.15	0.27

EC—electrical conductivity, OC—organic carbon, CaCO_3_—calcium carbonate, CEC—cation exchange capacity, Na^+^—extractable sodium. Values are means ± standard error (*n* = 9), and letters indicate statistically significant differences between treatment means identified by Fisher’s least-significance difference test (LSD_0.05_). The levels of significance were represented by *p* > 0.05: *p* < 0.001 (***).

**Table 2 plants-13-03222-t002:** Total and water-soluble concentrations of potentially harmful elements (PHEs) in mg kg^−1^ in recovered soil (RS), polluted soil (PS), and Technosols (TO, TS, and TV).

	Total (T)						Water-Soluble (W)					
Soils	As_T_	Cd_T_	Cu_T_	Fe_T_ *	Pb_T_	Zn_T_	As_W_	Cd_W_	Cu_W_	Fe_W_	Pb_W_	Zn_W_
**RS**	51.44 ± 0.63 a	1.98 ± 0.05 a	112.03 ± 5.12 b	29.88 ± 0.25 a	117.36 ± 1.01 a	420.35 ± 4.37 d	0.269 ± 0.015 c	0.010 ± 0.001 a	0.405 ± 0.015 b	9.943 ± 0.706 c	0.078 ± 0.010 b	0.606 ± 0.034 b
**PS**	345.79 ± 5.64 d	7.85 ± 0.13 d	241.34 ± 5.60 d	52.98 ± 0.79 c	545.86 ± 9.05 d	189.14 ± 2.40 b	0.135 ± 0.009 a	0.101 ± 0.010 b	2.617 ± 0.038 d	3.209 ± 0.125 a	bdl a	7.061 ± 0.098 e
**TO**	207.84 ± 1.85 c	4.21 ± 0.27 b	99.65 ± 1.19 ab	38.25 ± 0.64 b	329.30 ± 5.85 c	163.07 ± 1.73 a	0.377 ± 0.013 d	0.015 ± 0.001 a	3.301 ± 0.053 e	7.718 ± 0.409 b	0.089 ± 0.007 b	1.585 ± 0.043 d
**TS**	178.71 ± 3.75 b	5.86 ± 0.13 c	137.63 ± 3.06 c	38.29 ± 0.39 b	281.54 ± 4.90 b	315.76 ± 5.71 c	0.457 ± 0.014 e	0.014 ± 0.001 a	0.696 ± 0.015 c	19.971 ± 1.010 d	0.046 ± 0.006 a	0.913 ± 0.019 c
**TV**	212.20 ± 1.76 c	5.88 ± 0.09 c	93.70 ± 1.95 a	39.91 ± 0.54 b	333.45 ± 3.44 c	181.46 ± 2.41 b	0.199 ± 0.012 b	0.006 ± 0.001 a	0.130 ± 0.007 a	3.251 ± 0.194 a	bdl a	0.321 ± 0.018 a
***p*-value**	***	***	***	***	***	***	***	***	***	***	**	***
**LSD_0.05_**	9.29	0.44	12.86	1.58	15.82	10.41	0.04	0.01	0.09	1.69	0.02	0.15

Values are means ± standard error (*n* = 9), and letters indicate statistically significant differences between treatments means identified by Fisher’s least-significance difference test (LSD_0.05_). The levels of significance are represented by *p* > 0.05: *p* < 0.01 (**), and *p* < 0.001 (***). * in g kg^−1^. bdl: below the detection limit.

**Table 3 plants-13-03222-t003:** Concentrations of potentially harmful elements (PHEs) in mg kg^−1^ in aerial part and roots of *L. sativa* plants grown in recovered soil (RS), polluted soil (PS), and Technosols (TO, TS, and TV).

	Aerial Part						Roots					
Soils	As	Cd	Cu	Fe	Pb	Zn	As	Cd	Cu	Fe	Pb	Zn
**RS**	2.21 ± 0.22 b	0.88 ± 0.03 c	57.14 ± 0.79 a	85.80 ± 7.14 a	bdl a	55.37 ± 0.46 b	5.10 ± 0.86 b	0.97 ± 0.04 a	101.40 ± 2.81 ab	5587 ± 132 a	6.51 ± 0.48 b	81.89 ± 6.08 a
**PS**	0.65 ± 0.35 a	0.55 ± 0.02 b	152.59 ± 1.08 b	70.32 ± 13.79 a	0.78 ± 0.26 ab	129.31 ± 3.21 d	0.78 ± 0.68 a	4.51 ± 0.72 b	225.53 ± 6.00 c	6074 ± 139 a	2.40 ± 0.30 a	893.85 ± 23.45 b
**TO**	1.79 ± 0.14 ab	0.27 ± 0.01 a	185.56 ± 3.12 c	73.98 ± 7.04 a	0.38 ± 0.14 a	49.53 ± 0.28 ab	18.23 ± 0.72 d	0.95 ± 0.81 a	87.58 ± 9.39 a	7322 ± 313 b	11.28 ± 0.42 c	65.27 ± 3.36 a
**TS**	1.55 ± 0.18 ab	0.23 ± 0.04 a	363.08 ± 7.08 d	164.51 ± 12.10 b	1.32 ± 0.41 b	44.06 ± 1.03 a	17.15 ± 0.68 cd	0.83 ± 0.68 a	101.62 ± 7.07 ab	8003 ± 628 b	13.12 ± 0.49 d	63.53 ± 5.03 a
**TV**	2.56 ± 0.17 b	1.57 ± 0.02 d	68.86 ± 1.53 a	67.01 ± 6.71 a	bdl a	92.97 ± 1.76 c	14.90 ± 0.81 c	1.40 ± 0.81 a	112.82 ± 2.07 b	7741 ± 113 b	19.88 ± 0.34 e	83.87 ± 1.12 a
***p*-value**	*	***	***	***	**	***	***	***	***	***	***	***
**LSD_0.05_**	1.12	0.08	11.82	28.10	0.62	5.42	2.18	2.24	17.10	883.71	1.11	31.84

Values are means ± standard error (*n* = 9), and letters indicate statistically significant differences between treatments means identified by Fisher’s least-significance difference test (LSD_0.05_). The levels of significance were represented by *p* > 0.05: *p* < 0.05 (*), *p* < 0.01 (**), and *p* < 0.001 (***). bdl: below the detection limit.

**Table 4 plants-13-03222-t004:** Photosynthetic and gas exchange parameters of *L. sativa* plants grown in recovered soil (RS), polluted soil (PS), and Technosols (TO, TS, and TV).

Soils	ACO_2_ (µmol CO_2_ m^−2^ s^−1^)	E (mol H_2_O m^−2^ s^−1^)	g_s_ (mmol CO_2_ m^−2^ s^−1^)	C_i_ (µmol CO_2_ mol air^−1^)	WUE (µmol CO_2_ mol H_2_O^−1^)
**RS**	4.34 ± 0.15 c	1.21 ± 0.05 d	93.40 ± 3.62 d	311.63 ± 0.77 c	3589.78 ± 30.19 a
**PS**	0.98 ± 0.09 a	0.16 ± 0.01 a	11.50 ± 0.63 a	257.85 ± 7.80 b	5827.05 ± 338.38 b
**TO**	2.52 ± 0.19 b	0.80 ± 0.07 c	59.40 ± 5.22 c	313.34 ± 5.82 c	3436.03 ± 256.61 a
**TS**	2.89 ± 0.09 b	0.37 ± 0.01 b	26.80 ± 0.89 b	215.43 ± 4.18 a	7836.31 ± 196.14 c
**TV**	4.38 ± 0.14 c	1.26 ± 0.03 d	95.00 ± 3.36 d	311.94 ± 0.08 c	3491.64 ± 42.17 a
***p*-value**	***	***	***	***	***
**LSD_0.05_**	0.36	0.11	8.60	12.82	563.33

Net photosynthetic rate (ACO_2_), transpiration rate (E), stomatal conductance (g_s_), intracellular CO_2_ concentration (C_i_), and water use efficiency (WUE). Values are means ± standard error (*n* = 9), and letters indicate statistically significant differences between treatment means identified by Fisher’s least-significance difference test (LSD_0.05_). The levels of significance were represented by *p* > 0.05: *p* < 0.001 (***).

## Data Availability

The data presented in this study are available from the corresponding author on request.

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
