# Peer review of "Physiological Response of Lettuce (*Lactuca sativa* L.) Grown on Technosols Designed for Soil Remediation"

_plants, 2024, doi:10.3390/plants13223222_

Round 1

Reviewer 1 Report

Comments and Suggestions for Authors

The topic presented in this manuscript is indeed of interest and well suits to the journal objective. The work is generally well presented and several results covering different aspects of the work are reported. Some further improvements are still required. As follows the comments to be addressed:  

- Introduction section: The overall section is quite long and many  information about the topic, despite important, could also be condensed. Furthermore, Authors lack to stress out the novelty of their work which maybe results moved to the background due to the lengthiness of the section. 

- Page 2, line 90: "...of H2O2...", these should be reported as subscripts

- Page 7, line 258: "...to slightly alkaline pH (SR...", what is the acronym "SR"?

- Page 9, line 346: "... and SR...", also here is reported a wrong acronym.

- Materials and Methods section: Materials and methods section should be added before the results in order to provide a logical thread in the manuscrip organization.

- Page 11, line 402: "Several remediation actions were implemented...", Authors should briefly provide some more detail about this. Which remediation actions were carried out?

- Page 11, line 406: "...at five different locations...", these are not really clear from Figure S1. From the second Figure only four circled areas can be observed. Moreover, sampling point coordinates added to the Figure caption should be added. 

- Page 13, line 483: "The instrument was heated for 30 min...", at which temperature? 

- Conclusions section: Authors could provide some further perspective worth of further investigation about this topic.  

Comments on the Quality of English Language

The English language is clear and quite fine. Authors should check for some minor error throughout the manuscript. 

Author Response

Attached document

Reviewer 2 Report

Comments and Suggestions for Authors

The study assesses the physiological response of lettuce plants (Lactuca sativa L ) grown on Technosols for the remediation of polluted soils containing harmful elements (PHEs: As, Cd, Cu, Fe, Pb, Zn). To this purpose, soil properties and PHEs solubility were investigated, together with physiological parameters, photosynthetic capacity, oxidative stress and antioxidant defence to evaluate the phytotoxicity of the elements. The results display a reduction in PHEs-phytotoxicity suggesting this kind of eco-technology as possible solution for soil remediation purposes, with effectiveness of Technosols depending on its organic components. In particular, the Technosol composed of pruning and gardening vermicompost (TV) showed an overall performance far superior to that of the other Technosols. The study is significant for the ‘remediation research field’.

I strongly recommend providing a graphical representation of the treatments, as there are many. For example, you could create a graphical scheme describing the treatments listed in lines 14-17 and 410-416. 

Table 4: please, specify the unit of measure. For instance, µmol of what? mmol of what?

Tables and figures display the level of significance. However, statistical significance differences should also be indicated using different letters to highlight the differences among treatments RS, PS, TO, TS, and TV.

Lines 411-416: Why Technosols were designed and produced by mixing the components in the indicated ratios? What is the reason for mixing 60% of polluted soil (PS) with 18% of organic-rich waste (olive-mill by-product, composted sewage sludge or vermicompost from pruning and gardening waste), 20% of marble cutting and polishing sludge, and 2% of iron oxyhydroxide-rich surge [IO].

Lines 433-434: ‘Seven days after transplanting, acute phytotoxic effects began to be observed in some treatments, thus ten days later, all L. sativa plants from each treatment were collected’. What kind of acute phytotoxic effects? Please, specify

Lines 490-507: did you use standards for these determinations? Provide details

Introduction and Discussion: In the recent paper doi: 10.1007/s11252-024-01552,  researchers evaluated the possibility of growing herbaceous, shrubby, and tree plant species on Technosol for greening purposes. Notably, the application of plant-based compost positively influenced both the photosynthetic activity and overall growth of these plants in the long term. This study may serve as a valuable resource for the introduction and discussion of data related to the rehabilitation of degraded areas and ecological restoration.

Author Response

Attached document

Round 2

Reviewer 1 Report

Comments and Suggestions for Authors

The Authors provided proper revisions and improved their manuscript.

Accordingly, the manuscript can be considered for acceptance.

Author Response

Thank you for time and effort.